# Screening for Subclinical Atherosclerosis and the Prediction of Cardiovascular Events in People with Type 1 Diabetes

**DOI:** 10.3390/jcm13041097

**Published:** 2024-02-15

**Authors:** Tonet Serés-Noriega, Verónica Perea, Antonio J. Amor

**Affiliations:** 1Diabetes Unit, Endocrinology and Nutrition Department, Hospital Clínic, 08036 Barcelona, Spain; 2Endocrinology and Nutrition Department, Hospital Universitari Mútua de Terrassa, 08221 Terrassa, Spain

**Keywords:** type 1 diabetes, cardiovascular risk, atherosclerosis, subclinical atherosclerosis, plaque, carotid plaque, coronary calcium score, coronary computed tomography angiography, magnetic resonance, ankle–brachial index

## Abstract

People with type 1 diabetes (T1D) have a high cardiovascular disease (CVD) risk, which remains the leading cause of death in this population. Despite the improved control of several classic risk factors, particularly better glycaemic control, cardiovascular morbidity and mortality continue to be significantly higher than in the general population. In routine clinical practice, estimating cardiovascular risk (CVR) in people with T1D using scales or equations is often imprecise because much of the evidence comes from pooled samples of people with type 2 diabetes (T2D) and T1D or from extrapolations of studies performed on people with T2D. Given that T1D onsets at a young age, prolonged exposure to the disease and its consequences (e.g., hyperglycaemia, changes in lipid metabolism or inflammation) have a detrimental impact on cardiovascular health. Therefore, it is critical to have tools that allow for the early identification of those individuals with a higher CVR and thus be able to make the most appropriate management decisions in each case. In this sense, atherosclerosis is the prelude to most cardiovascular events. People with diabetes present pathophysiological alterations that facilitate atherosclerosis development and that may imply a greater vulnerability of atheromatous plaques. Screening for subclinical atherosclerosis using various techniques, mainly imaging, has proven valuable in predicting cardiovascular events. Its use enables the reclassification of CVR and, therefore, an individualised adjustment of therapeutic management. However, the available evidence in people with T1D is scarce. This narrative review provides and updated overview of the main non-invasive tests for detecting atherosclerosis plaques and their association with CVD in people with T1D.

## 1. Introduction

Cardiovascular disease is the leading cause of mortality and morbidity worldwide. Although medical advances have reduced the incidence of death over the past decade, global prevalence and mortality have continued to rise [1]. Atherosclerotic cardiovascular disease (ASCVD), mainly coronary artery disease (CAD) and atherothrombotic stroke, represents the leading cause within this group, accounting for over 13 million deaths in 2021 [2].

CVR is often assessed using various equations that estimate the 10-year probability of suffering an event. This strategy has several disadvantages: (1) most of them are not applicable in people aged <40 years, and age greatly influences the estimated risk, which makes it difficult to identify young people at high risk; (2) they are designed to use cross-sectional data, although the impact of the main risk factors occurs cumulatively; and (3) they consider only a few classical risk factors (e.g., sex, age, smoking habit, cholesterol levels or systolic blood pressure) and leave it to the clinician to decide how to weight the risk indicated by several other variables that modify CVR (e.g., social deprivation, the presence of obstetric factors or autoinflammatory diseases). Furthermore, there is no clear consensus between the different strategies. Several studies show heterogeneous recommendations depending on the equation used, and a lower than desired discriminatory power [3,4,5]. All this underlines the need for additional tools to better assess CVR in each individual.

The prelude to an acute event is the formation of atheromatous plaques in large and medium-sized arteries, which begins early in life and progresses silently over several years. The detection of subclinical atherosclerosis, which can be easily assessed using mostly non-invasive imaging tests, is one of the main strategies employed to individualise this risk. The presence of atheromatous plaques is associated with incident cardiovascular events in studies on large population cohorts without diabetes [6,7,8]. It allows us to identify young individuals at high CVR who may benefit from a long-term preventive strategy, knowing that they could benefit most from CVR control such as low-density lipoprotein cholesterol (LDLc) levels or blood pressure [9,10]. This is important because statins [11] and PCSK9 inhibitors [12,13] can delay the process of atherosclerosis, stabilise plaques already formed and reduce the likelihood of cardiovascular events. In this sense, the main clinical guidelines for cardiovascular prevention consider subclinical atherosclerosis detection as a risk-modifying factor, primarily in intermediate or borderline risk patients, both up- and down-regulating, with the consequent changes in treatment and follow-up that this entails [14,15]. In addition, the visualisation of atherosclerotic plaques by patients themselves is not only useful for the clinician but can also improve adherence to lifestyle measures and treatments with proven cardioprotective effects (e.g., lipid-lowering, antihypertensive and antiplatelet therapy) [16].

People with T1D have a four to eight times higher risk of CVD than the general population [17,18]. The physiopathology of T1D is characterised by the rapid and early autoimmune destruction of pancreatic beta cells, resulting in hyperglycaemia and the requirement for lifelong insulin replacement therapy. Hyperglycaemia is one of the most important CVR factors; however, even those with optimal glycaemic control (time-updated haemoglobin A1c (HbA1c) ≤ 6.9% or 51.9 mmol/mol) have a three-fold increased risk of CVD death compared with their counterparts without diabetes [19]. This fact suggests the existence of other factors involved in the pathogenesis of CVD in T1D such as exposure to hypoglycaemia, glycaemic variability, quantitative and qualitative abnormalities of lipoproteins, immune dysfunction, inflammation or cardiac autoimmunity, among others [20,21]. Furthermore, a recent Mendelian randomisation study supports the hypothesis of the presence of T1D as a CVD causal factor [22]. Previously, we showed how equations for estimating CVR that are not specific to T1D can have poor diagnostic performance [23]. Although several methods have been developed for this purpose [24,25], most assessments of CVR and cardiovascular management are based on extrapolations from studies in people with T2D. The two entities have different pathophysiologies. They share little beyond hyperglycaemia. These differences highlight the many mechanisms in the mechanisms leading to CVD. It is essential to develop reliable strategies to classify CVR and implement preventive strategies early in this vulnerable population.

Against this background, this narrative review aims to evaluate the clinical utility of using non-invasive techniques to detect subclinical atherosclerosis in people with T1D. Due to their invasiveness, cost and lower applicability for screening in clinical practice, diagnostic tests such as angiography, intravascular ultrasound and intravascular optical coherence tomography will not be reviewed in this manuscript.

## 2. Methods of Searching

In the present study, a narrative review of the literature was carried out focused on the different tools for the detection of atherosclerosis and its association with CVD in people with T1D. A comprehensive search of international the PubMed and Embase (Elsevier, Amsterdam, The Netherlands) databases was conducted for all articles available up to 12 January 2024. The suitability of articles collected from the electronic search was reviewed based on the abstracts. The search criteria were from lowest to highest specificity depending on the number of results available. We took all the results into account. We prioritised articles that evaluated clinical variables and were published in high-impact journals. Articles that were not related to the objective of the manuscript, conference abstracts, and duplicate articles were excluded from the review process. Only articles published in English were considered.

## 3. Physiopathology of Atherosclerosis in Diabetes

Atherosclerosis is a complex and not fully understood process. In brief, the evidence seems to state that it begins with the penetration and accumulation of apolipoprotein B-100-containing particles, mainly LDL particles, into the intimal layer of the arterial wall. In this new environment, they are oxidised and modified, leading to an inflammatory and immunogenic activation. Although the exact mechanisms are not fully understood, several processes, such as increased oxidative stress [26] and the degeneration of the endothelial glycocalyx [27], have been implicated. Subsequently, circulating T lymphocytes and monocytes enter the intimal layer through a dysfunctional endothelium. The latter mature into macrophages expressing scavenger receptors that recognise these modified lipoprotein particles and internalise them, notably increasing the cholesterol content of macrophages, turning them into foam cells. These foam cells release a plethora of proinflammatory cytokines that promote the process of atherosclerosis [28].

These leukocytes produce various mediators. The mediators cause smooth muscle cells to move from the media layer to the intima. There, the smooth muscle cells can grow and produce extracellular matrix molecules that increase the size of the plaque. From here, inflammation is perpetuated, and multiple processes occur that influence the progression of atherosclerosis. Finally, there are mainly two plaque complications: intraluminal growth with vascular stenosis and rupture or erosion with intravascular thrombus formation [29].

Various factors enhance and/or accelerate several of the above processes in people with diabetes. For example, hyperglycaemia leads to the glycation of various proteins, which undergo multiple reactions culminating in the formation of advanced glycation end products (AGEs). AGEs have been implicated in several steps in the development of atheromatous plaques, including accelerated monocyte migration, the glycation of lipoproteins facilitating the recognition by macrophages, an increased production of inflammatory cytokines and procoagulant effects, among others [30]. In addition, the increased productions of sorbitol and fructose (polyol pathway) also increase the production of AGEs. Further, AGEs are hardly degradable and may persist over time, which may explain why those who have had poor glycaemic control are at increased risk of vascular complications despite a better current control; this is known as the legacy effect [31,32].

Notwithstanding, through various mechanisms, mainly intracellular hyperglycaemia, there is an increase in oxidative stress with an increased production of reactive oxygen species, which react with various structures such as nucleic acids and proteins, increasing the expression of adhesion and inflammatory factors and affect genes involved in the pathogenesis of atherosclerosis [33,34]. There is an increased synthesis of pro-inflammatory cytokines in people with diabetes and several inflammatory markers have been associated with atherosclerosis in T1D [35,36,37]. In addition, elevated glucose levels and other processes increase the production of diacylglycerol, which, together with calcium, activates protein kinase C. Its activation has been implicated in several steps of atheroma plaque formation [30,34,38]. Furthermore, people with diabetes have qualitative and quantitative changes in the lipoprotein metabolism that have been implicated in the process of atherogenesis [39,40,41]. Most of the above processes are interrelated and promote each other. This perpetuates and accelerates the process of atherosclerosis.

Finally, people with diabetes not only have a greater atherosclerotic burden than the general population [42], but also have a greater inflammatory infiltrate, necrotic core and calcification that have been linked with increased plaque vulnerability [43]. Several of these changes are due to hyperglycaemia, but the high residual risk indicates that other agents are also involved. Even a recent Mendelian randomisation study suggests a possible causal role of T1D in peripheral and coronary atherosclerosis after adjusting for confounders (comorbidities, classic CVR factors, lipid and inflammatory variables), with a partial mediation of the effect through hypertension [22].

## 4. Screening Methods for Subclinical Atherosclerosis

### 4.1. Carotid Ultrasound

The carotid territories, and especially the carotid bifurcation, are prone to atherosclerosis in predisposed individuals with CVR factors due to physiological changes in blood flow. As this is a shallow area, ultrasound using high-frequency probes (usually > 9 MHz) allows for a detailed visualisation of the arterial wall in its entire extracranial extent. The thickness of the intima and media layers (intima–media thickness or IMT) is measured and, although various cut-off points have been proposed, the consensus is that plaque is defined as a focal structure that encroaches into the arterial lumen by at least 0.5 mm or 50% of the surrounding IMT value or has an IMT thicker than 1.5 mm [44]. If the plaque is of significant size, the degree of stenosis is estimated using a variety of methods, most notably through the use of flow velocities in the stenotic area using pulsed-wave Doppler mode. In addition to conventional ultrasound, other related tools include contrast-enhanced ultrasound (CEUS), which allows the characterisation of features associated with plaque vulnerability such as neovascularisation [45], and 3D volumetric ultrasound (3DVUS), which helps in the spatial interpretation and calculation of the total area of atheromatous lesions [46].

Unlike other methods, carotid ultrasound does not emit radiation, allows for the visualisation of plaques in the early stages (Figure 1), is inexpensive and quick to perform, and it is also common to have an ultrasound scanner in the office, enabling an easy assessment and the diagnosis and initiation of treatment at the same time. Focusing on the arterial wall rather than the lumen makes it easier to characterise plaques and identify those with features associated with a higher risk of complications such as echolucency or surface ulceration [47,48]. However, its main disadvantages are its operator dependency, which requires trained personnel, and the lack of standardisation of IMT measurements.

The usefulness of IMT measurement and the detection of carotid plaque in T1D has been poorly studied, and the results are controversial. Data from the Diabetes Control and Complications Trial/Epidemiology of Diabetes Interventions and Complications (DCCT/EDIC) cohort are consistent in that various CVRs such as age, male sex, systolic blood pressure, smoking, HbA1c and albuminuria are predictors of IMT progression and that intensive diabetic treatment slows such progression [49]. Subsequently, the association between measures of IMT and CAD events was analysed after 17 years of follow-up with more than 1300 subjects [50]. Increased common carotid artery IMT was consistently associated with CVD events incidence in models adjusted individually for age, sex, HbA1c, systolic blood pressure, HDL cholesterol, total cholesterol and smoking. However, despite a trend, this association did not remain significant after adjusting for all variables, nor did internal carotid artery IMT. It should be noted that they analysed baseline ultrasound data from a young sample (mean age 35 years) without information on atherosclerotic progression, which is, if anything, a more important measure than the mere presence of plaque [51]. In addition, the number of events was low (*n* = 135), which may affect the power of the study. However, carotid atherosclerosis in T1D is associated with many pathologies and alterations linked to a higher risk of CVD. These include high systolic blood pressure [52], preeclampsia [53,54], retinopathy [55,56], insulin resistance [57,58], excess weight gain [59], enlarged left ventricular mass [60], cerebral microbleeds [61], cognitive impairment [62], inflammation and endothelial dysfunction [36,63]. The aforementioned, in addition to the proven predictive capacity of CVD events in the general population [7,64], lead us to believe that, pending more properly designed prospective studies, there is no reason not to consider carotid ultrasound as a useful tool in the evaluation of these patients. Finally, the use of this technique is not limited to the phenotyping of CVD risk but may serve as a marker of treatment response [65,66,67] and may even lead the clinician to intensify cardioprotective treatment and the patient to adopt a healthier lifestyle and be more compliant with treatment [68].

### 4.2. Coronary Artery Calcium (CAC) Scan

When using non-contrast computed tomography (CT), the CAC of the epicardial arteries is measured, with an area of ≥ 1 mm^2^ and > 130 Hounsfield units being considered a lesion [69]. The product of the calcified area and the score of the lesion itself (ranging from 1 to 4) are known as the calcium score. The total CAC score is the sum of all calcium score units and is reported in Agatston units (AU), considering no CAC (0), minimal CAC (1–10), mild CAC (11–100), moderate CAC (101–400) and severe CAC (>400).

CAC scanning is a relatively fast, economical, reproducible and easy-to-interpret method. One of its limitations is that it emits radiation, albeit a small amount. In addition, it measures coronary calcification, an advanced stage of atherosclerosis, and misses obstructive and non-calcified lesions, the latter generally representing an earlier stage that may be present in young people with high CVR that might go unnoticed through this method. Also, statin treatment, despite its proven cardiovascular benefits, may lead to an increase in CAC due to the regression of non-calcified areas and promotion of intraplaque calcification [70,71], so it is unclear whether the CAC score may be useful as an additional tool for monitoring treatment. Calcifications are associated with higher plaque stability and lower event probability [72]. However, a higher CAC volume is positively associated with multiple CVR factors [73]. As a product that takes into account calcification and lesion area, an increased CAC score usually indicates a higher total atherosclerotic burden [74]. This is why it now stands as a powerful predictor of cardiovascular events [75].

In people with T1D, its usefulness as a predictor of CVD has been studied more than with other methods. In the DCCT/EDIC cohort [76], a consistent and proportional association between CAC score and CVD events was demonstrated after 10–13 years of follow-up. After adjusting for HbA1c and other CVR factors, a CAC score of > 100 AU was associated with subsequent CVD and major adverse cardiovascular events (MACE). However, it did not show a significant incremental value in ROC analyses after adding it to the model that already took into account the variables mentioned above. In the Pittsburgh Epidemiology of Diabetes Complications (EDC) study [77], similar results were observed. The baseline CAC score and CAC score progression 4–8 years later were independently associated with CVD events in childhood-onset T1D. The former showed increased predictive ability when associated with other established CVR factors. It should be noted that CAC score increases with age, with a positive CAC occurring in >80% of the general population aged 65 years or more [78]. The previously reported cohorts are from studies initiated more than 30 years ago that subscribe only to the USA, and the prevalence of a positive CAC score was 30–40% [76,77].

Data from different demographics, geographies and periods show widely differing prevalences, ranging from 18 to 94% [79,80,81,82], so studies assessing its predictive utility for CVD events in different settings are needed. In addition, it appears that people with T1D tend to have a greater progression of CAC compared to people without diabetes [83,84]. Data from the EDC study show that CAC progression increases with BMI, non-high-density lipoprotein cholesterol levels, the duration of diabetes and albuminuria [85]. From the same sample, it was published years later that baseline CAC and, above all, CAC progression were associated with cognitive impairment [86]. Similar results are also available from the Coronary Artery Calcification in Type 1 Diabetes (CACTI) study. CAC progression was associated with insulin sensitivity [87], nephropathy [88], menopause [84], markers of inflammation [89] and obesity [90]. CAC progression has also been associated with suboptimal glycaemic control [91]. Finally, intensive diabetes treatment in a DCCT trial was associated with lower CAC scores, an effect mainly mediated by the reduction in HbA1c during the study and the legacy effect seen 7–9 years later, when CAC was determined [92].

### 4.3. Coronary Computed Tomography Angiography (CCTA)

CCTA is a non-invasive technique using intravenous iodinated contrast. It allows the detection of CAD and the characterisation of atherosclerotic plaques in the epicardial arterial tree. After contrast administration, CT images are acquired during inspiratory breath hold to avoid artefacts resulting from chest motion [93]. To optimise the quality of the images, it is necessary not to exceed a certain pulsation threshold (usually < 60–65 beats/min), which sometimes requires the use of negative chronotropic drugs such as beta blockers, and makes their application difficult in the case of a high heart rate and/or non-sinus rhythm [94].

CCTA can be used as a prognostic tool to characterise calcified and non-calcified plaques, the latter of which may be present in young people and may even indicate greater plaque vulnerability and risk of complications [95,96]. It is also indicated in symptomatic patients, as one of its main advantages is that it detects obstructive lesions and allows the measurement of the stenosis degree. It is a reliable and reproducible technique, easy to interpret, and its results can be used to decide whether an invasive or therapeutic approach is required. Among its drawbacks is being a more costly, laborious technique, with radiation and contrast exposure. The use of iodinated contrast may limit its use in cases of nephropathy and/or allergy. Further, calcified lesions may incur false positives due to blooming artefacts. Also, image quality may be suboptimal in cases of obesity [94].

The FACTOR-64 clinical trial [97] evaluated the usefulness of screening with CCTA versus standard national guidelines-based optimal diabetes care in 900 people with diabetes (12% with T1D) and no symptoms of CAD. Based on the screening results, the decision was made to use standard, aggressive therapy with more adjusted control targets for various CVR factors or aggressive therapy associated with invasive coronary angiography, subsequently assessing the incidence of MACE. After 4 years of follow-up, although there was a trend in favour of the use of CCTA, it was not significant compared to the control group. It should be noted that (1) the number of events in both groups was lower than expected and (2) the control group had a good management of several CVR factors (baseline mean LDLc 87.7 mg/dL and mean systolic blood pressure 130.5 mmHg). Both of these points could have compromised the power to detect differences between the groups. There are no other studies available that assess CCTA as a screening test in this population; however, some indirect information is available. For example, a prospective study on over 2000 patients showed an association between atherosclerotic lesions detected by CCTA and several specific CVR factors such as glycaemic control, diabetic neuropathy or repeated hypoglycaemias [98]. In individuals with long-standing T1D, it has also been associated with LDLc levels [99,100] and glycaemic control [100]. Finally, it should be emphasised that in young adults, the identification of subclinical atherosclerosis is significantly more likely using CCTA compared to the CAC score, enabling earlier detection and intervention [101].

### 4.4. Ankle–Brachial Index (ABI)

The ankle–brachial index (ABI) estimates the presence of ischaemia in the lower extremities. It is a screening method for peripheral arterial disease (PAD) that provides indirect evidence of the presence of systemic atherosclerosis. According to the European Society of Cardiology (ESC) Peripheral Arterial Disease guidelines [102], it is performed in the supine position, with a cuff placed just above the ankle. After a 5–10-min rest, the systolic blood pressure is measured with a Doppler probe (5–10 MHz) in the posterior and the anterior tibial (or dorsalis pedis) arteries of each foot and on the brachial artery of each arm. The ABI of each leg is calculated by dividing the highest ankle SBP by the highest arm SBP. A value < 0.9 indicates ischaemia; between 0.9 and 1 is considered borderline; between 1 and 1.4, normal; and >1.4 is indicative of arterial stiffness or calcification.

Its main advantages are that it is inexpensive, standardised, easy to interpret and quick to perform. Yet, its results are operator-dependent and, as it does not directly visualise atherosclerosis, it does not allow the characterisation and quantification of plaques. In addition, it detects disease with haemodynamic repercussions and therefore cannot have sufficient sensitivity for early subclinical stages. A low performance has also been described in the case of arterial calcification, a fairly common occurrence in people with diabetes [103], and in haemodialysis patients in whom other alternatives such as the toe–brachial index could offer better results, as the measurements are performed on digital vessels that rarely suffer calcifications [104,105].

It should be noted that PAD is one of the most frequently occurring CVDs in people with T1D [18,106]. In screening for this complication, the use of ABI is recommended in the presence of consistent signs and symptoms [107]. ABI’s sensitivity is questionable in an asymptomatic population and/or at early stages of the disease [108,109]. Several meta-analyses associate high and low ABI values with CVD in the general population [110,111,112] and in people with T2D [113,114], although studies on T1D are scarce. In a cross-sectional study on 289 adults with T1D without PAD symptoms, an ABI < 0.9 was detected in 6% and ABI > 1.2 in 26%. Of those with abnormal ABI, 15% had ultrasound-assessed carotid atherosclerosis, and 40% had silent PAD confirmed through lower extremity arterial Doppler ultrasound and/or the toe–brachial index [115]. Another cross-sectional study conducted in 185 adults with T1D from the EDC study analysed the association between ABI scores and medial arterial calcification (MAC) assessed using lower extremity radiographs, as a proxy CVD. In total, 57% had MAC, 8% had ABI > 1.3 and another 8% had an ankle–brachial difference (ABD) >75 mmHg. The predictive ability of MAC using the ABI or ABD was modest, suggesting a higher diagnostic yield of ABD as opposed to ABI. However, in using these cut-off points, more than 40–50% of cases with MAC remained undiagnosed [116]. In summary, the designs of the above studies yielded results that should be considered exploratory, and studies analysing the incidence of CVD based on ABI results in the asymptomatic T1D population are required.

### 4.5. Magnetic Resonance Imaging (MRI)

Magnetic resonance imaging (MRI) includes many techniques used in different vascular beds and provide information on both morphology and functionality (mainly cardiac). Its excellent resolution in soft tissues enables the distinction of plaque features (Table 1). The use of intravenous contrast with gadolinium is common and permits the assessment of neovascularisation and other vulnerability traits such as intra-plaque haemorrhage or fibrous cap thickness. One of its main drawbacks is that it is susceptible to motion-related artefacts (e.g., cardiac and respiratory movements). Therefore, its main application is in large vessels (e.g., carotid arteries or the aorta), which are relatively immobile. Its use in smaller vessels such as coronary arteries is complex, although advances in motion artefact correction, image acceleration and reconstruction techniques have improved [117]. It is also useful as a myocardial perfusion study as it has a higher spatial resolution than classical radionuclide studies [118]. In general, despite the versatility of MRI, its widespread clinical application is hampered by extended acquisition times, intricate scan planning and cost.

In this regard, MRI’s use as a screening method in the T1D population is infrequent. Weckbach et al. performed whole-body MRI and MR-angiography on 65 patients with diabetes duration >10 years (31% with T1D) and 200 healthy individuals and observed a markedly higher prevalence of atherosclerosis in almost all territories studied in those with diabetes. Of note, >50% of clinically significant changes (e.g., carotid, vertebral or renal artery stenosis, pattern compatible with myocardial infarction or cerebral ischaemic infarction) were previously unknown, suggesting the usefulness of using their screening protocol in this population [119]. Further, a cross-sectional study in 136 adult patients with long-standing T1D suggested a higher plaque burden in the right coronary artery in those with albuminuria >300 mg/24 h. There were no differences when assessing the abdominal and thoracic aorta [120]. The results of the above studies are in line with expectations and evidence from other screening tests for subclinical atherosclerotic disease.

### 4.6. Other Screening Methods

The main non-invasive tools used in clinical practice for the detection of atherosclerotic plaques have been described. However, there are other available techniques that are either novel or do not directly study atherosclerosis itself. For example, increased arterial stiffness or arteriosclerosis, a different process but often associated with atherosclerosis, has been associated with micro- and macrovascular complications in T1D [121]. Furthermore, molecular imaging using positron emission tomography (PET), PET-CT, PET-MRI or nuclear MRI are promising tests that allow the study of multiple early processes involved in atherosclerosis using radiotracers (e.g., targeting different markers of inflammation, extracellular matrix components or macrophages) [37,122,123,124]. Finally, perfusion and functional cardiac imaging could enhance the accuracy of detecting CAD in subjects with high CVR, such as people with T1D. Resting global longitudinal strain has also been highlighted as an early marker of myocardial damage in people with chronic coronary syndromes, and stress echocardiography has been used to detect coronary artery disease burden [125,126].

## 5. Future Directions

The current healthcare system is mainly focused on the advanced stages of atherosclerosis, showing that we act “too late and too little” in the early stages. Atheroma plaque formation starts in childhood and adolescence and is consistently associated with the presence of multiple modifiable CVRs [127]. The Progression of Early Subclinical Atherosclerosis (PESA) study showed that the prevalence of subclinical atherosclerosis was > 60% in middle-aged men, lagging behind the disease (atherosclerosis) by that time [128]. In people with T1D, the need for biomarkers to identify those most likely to benefit from pharmacological and non-pharmacological treatments with proven cardiovascular benefits becomes even more apparent.

The ideal screening method should detect atherosclerotic lesions in their earliest stages. It should also reliably predict cardiovascular events. It should also be efficient and applicable regularly to assess CVR longitudinally and dynamically. A tool that meets all these requirements is not yet available or it does not have the scientific evidence to support its usefulness with certainty. Anyway, given the pathophysiological complexity of CVD and the multiple factors involved, the optimal approach probably lies in the combined use of imaging techniques, the assessment of classical risk factors (e.g., conventional lipid profile, high blood pressure), polygenic scoring and tests aimed at detecting other alterations associated with early atherosclerosis (e.g., endothelial dysfunction, inflammatory markers [e.g., C-Reactive Protein, IL-1, IL-6, IL-18], proteomic or transcriptomic analyses and leptin and ghrelin dysregulations) (Figure 2). Advances must be aimed not only at detecting the earliest alterations of atherosclerotic lesions but also at identifying subjects at greater risk of plaque development, progression and acute complications. Longitudinal studies are needed to assess an appropriate screening strategy in these cases. Pending further and better evidence, in light of what is available, we advocate for early and systematic screening for subclinical atherosclerosis in this vulnerable population.

## 6. Conclusions

Little evidence is available on the usefulness of screening for subclinical atherosclerosis in the T1D population. So far, studies suggest similar results to those found in the general population. The use of the CAC score is probably the best predictor of future cardiovascular events, especially CAD. However, since it detects lesions at a more advanced stage, it is not useful in the younger population. The use of other techniques such as carotid ultrasound, CCTA and MRI may be useful in these earlier stages of atherosclerosis; however, more studies are needed to recommend their routine use as a screening test in T1D. Finally, ABI is useful as a first diagnostic test for PAD, but its ability to detect lesions without haemodynamic impact is low.

## Figures and Tables

**Figure 1 jcm-13-01097-f001:**
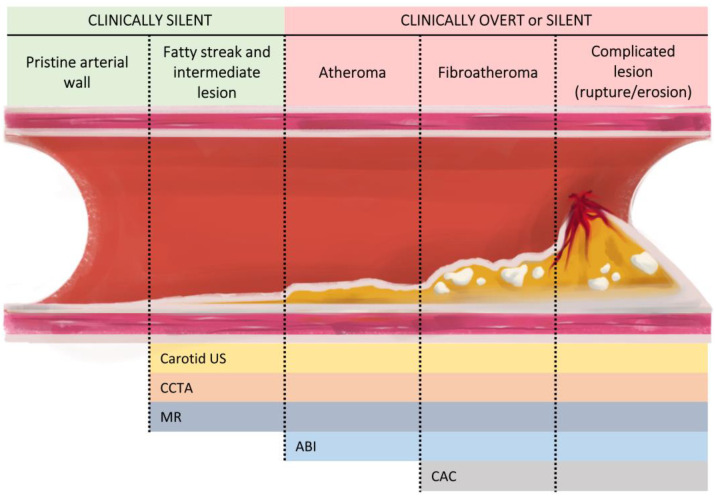
Main imaging screening methods according to atherosclerosis progression. The progression of atherosclerosis begins with the formation of lesions with fatty streaks that progress to intermediate lesions that are clinically asymptomatic but already detectable by various imaging methods. It then progresses to atheroma plaque, which, depending on the degree of stenosis and haemodynamic impact, may yield a pathological ABI result. Over time, the plaques increase their fibrotic content and may present calcifications. At this point, the use of CAC is sufficiently sensitive for their detection. Finally, atherosclerotic lesions may become complicated through rupture or erosion and lead to acute cardiovascular events. ABI: ankle–brachial index; CAC: coronary artery calcium; CCTA: coronary computed tomography angiography; MR: magnetic resonance; US: ultrasound.

**Figure 2 jcm-13-01097-f002:**
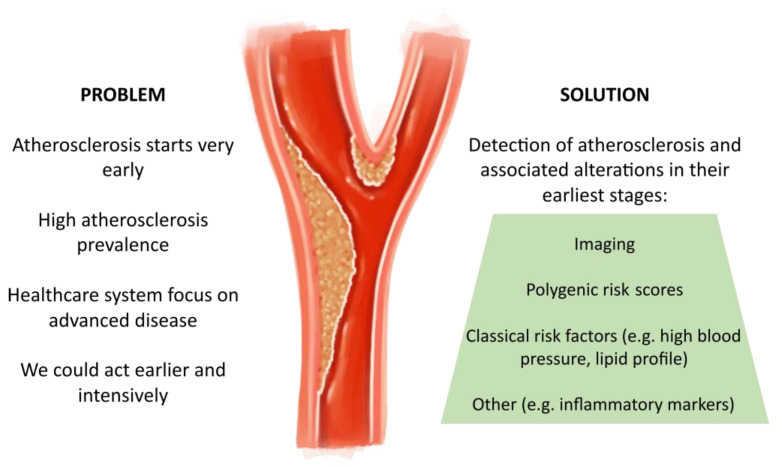
Future directions.

**Table 1 jcm-13-01097-t001:** Characteristics of the main non-invasive imaging methods for detecting subclinical atherosclerosis.

Imaging Modality	Strengths	Limitations
Carotid US	Identifies atherosclerosis in early stages Enables the differentiation of some plaque characteristics (better characterisation if CEUS or 3DVUS is used) Low cost No radiation exposure	Operator-dependent Lack of methodological standardisation of IMT measurements
CAC	Reproducible and standardised Strong CVD predictor Low cost	Radiation exposure (low) Does not identify plaque in early stages (only detects calcified plaques) Doubtful usefulness for monitoring treatment (promotion of plaque calcification with statin use)
CCTA	Reproducible and standardised Detects obstructive coronary lesions and degree of stenosis (direct detection of CAD) Plaque characterisation	Expensive Radiation exposure Iodinated contrast exposure (allergy, nephrotoxicity) Imaging is limited if arrhythmias and/or obesity Blooming artefact in calcified lesions
ABI	Standardised Easy-to-interpret results Inexpensive No radiation exposure Quick to perform	Does not visualise atherosclerosis Primarily picks up hemodynamically significant lesions and therefore may miss subclinical disease Limited interpretation in certain patient populations (high calcification burden, haemodialysis)
MR	Excellent soft tissue resolution for plaque characterisation Multiparametric (morphology, functionality) No radiation exposure	Motion artefacts (mainly limited to large-calibre vessels) Expensive Time consuming

ABI: ankle–brachial index; CAC: coronary artery calcium; CAD: coronary artery disease; CCTA: coronary computed tomography angiography; CEUS: contrast-enhanced ultrasound; CVD: cardiovascular disease; IMT: intima-media thickness; MR: magnetic resonance; US: ultrasound; 3DVUS: 3D vascular ultrasound.

## Data Availability

Not required as it is a narrative review. All the data is in the references.

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
