# Peer review of "Screening for Subclinical Atherosclerosis and the Prediction of Cardiovascular Events in People with Type 1 Diabetes"

_jcm, 2024, doi:10.3390/jcm13041097_

Round 1
Reviewer 1 Report
Comments and Suggestions for Authors
JCM-Manuscript ID-2856833
Reviewer Report
Thank you for giving me a chance to review this study. It is a good study
Screening for subclinical atherosclerosis and prediction of cardiovascular events in people with type 1 diabetes
Dear author,
This manuscript is well designed and well-presented and it is interesting study in current society. Against this background, this narrative review aims to evaluate the clinical utility of using non-invasive techniques to detect subclinical atherosclerosis in people with T1D. There are some corrections that the authors need to address first.
Comments
1. Abstract: Very less content. Kindly increase to 250 to 300 words.
2. Main manuscript:
3. Introduction: Since you are talking about the Type I Diabetes Mellitus it is insulin dependent diabetes mellitus. So please write the relationship or connection between the Cardiovascular risk with Type I DM.
4. Explain in detailed manner in Introduction about Type I DM..
5. Why you focused on TYPE I DM? Why not Type II DM. Justify………
6. Section 2. Physiopathology of atherosclerosis in diabetes Line no 83, 84 and 85. Authors mentioned about scavenger receptor. If you provide more details or role of scavenger receptor in pathophysiology from text books or articles.. it would benefit more
The latter mature into macrophages expressing scavenger receptors that allow the binding of lipoprotein particles and transform into foam cells.
7. I would request to include invasive test together in this review article, example cholesterol profile and CRP, Leptin and Ghrelin level in serum, IL-1, IL-6 and IL-18. ( Total cholesterol, LDL, HDL, VLDL and Triglycerides etc) this would benefit this article more and because it is a narrative review….
8. Method section is not there. Methods: Please include method. Mention the year of study and duration of article searched, language, filters used.
I suggest the authors to kindly incorporate the above comments…
Good Luck
Reviewer 2 Report
Comments and Suggestions for Authors
Cardiovascular diseases in diabetic patients are serious complications and early detection often proves to be a boon. In the current Review article by Ser’es-Noriega et al, the authors have discussed the methods/ tests used to screen the subclinical atherosclerosis and prediction of cardiovascular diseases in patients with type 1 diabetes. The review primarily focuses on carotid ultrasounds, coronary artery calcium scan, coronary computed tomography angiography, ankle-brachial index and magnetic resonance imaging.
In the current comprehensive review, the authors have satisfactorily provided a summary on the techniques along with strengths and limitations of each technique. All the subsections are well-written and provided information and statements are supported by literature. Table 1 is informative and summarizes the paper.
Author Response
We appreciate the reviewer's comment.
Reviewer 3 Report
Comments and Suggestions for Authors
This manuscript by Tonet Serés-Noriega et al. evaluated non-invasive tests for detecting atherosclerotic plaques and their correlation with CVD in individuals with T1DM.
Congratulations to the authors for this intriguing literature review.
There are some issues to consider:
- The type of revision (narrative) must be specified even in the abstract.
- Lines 260-261: I suggest modifying the sentence as CCTA is recommended in low or intermediate-risk patients according to the ESC guidelines on CCS.
- In the paragraph "Other screening methods," I suggest considering and discussing the role of strain echocardiography in detecting subtle CAD (according also to Duke score) in CCS patients (e.g., “Resting global longitudinal strain and stress echocardiography to detect coronary artery disease burden”) and mention that perfusion and functional imaging could enhance accuracy in detecting CAD in high-risk patients, such as those with T1DM (e.g. 10.1016/j.ijcard.2022.07.038; 10.3390/diagnostics13122083).
- Lastly, Figure 1 is very well done. I recommend creating a central illustration based on the content of the "Future directions" paragraph.
Minor English revision and double-checking abbreviations is required.
Round 2
Reviewer 3 Report
Comments and Suggestions for Authors
Congratulations to the authors for the significant improvements made in this revision, enhancing the overall quality of the manuscript. Additionally, kudos for the inclusion of valuable figures that contribute to the clarity and depth of the content.
Comments on the Quality of English LanguageMinor English revision is recommended.
